



# Silicon cycled by tropical forest trees: effects of species, elevation and bedrock on Mount Kinabalu, Malaysia

Ryosuke Nakamura[*,1], Hidehiro Ishizawa[2, 3], Rota Wagai[4], Shizuo Suzuki[1,5], Kanehiro Kitayama[1], and Kaoru Kitajima[1]

[1] Graduate School of Agriculture, Kyoto University, Kyoto, Japan
[2] Faculty of Agriculture, Kyoto University, Kyoto, Japan
[3] Graduate School of Engineering, Osaka University, Osaka, Japan
[4] National Agriculture and Food Research Organization, Institute for Agro-Environmental Sciences, Tsukuba, Ibaraki, Japan
[5] National Institute of Technology, Numazu College, Numazu, Shizuoka, Japan

*Correspondence to*: Ryosuke Nakamura (rnakamura3825@gmail.com)

**Abstract.** Plant species differ in degrees of silicon (Si) uptake and accumulation, and may differentially influence biogeochemical cycles of Si, possibly in interaction with other environmental factors. Here, we report how patterns of Si cycling by vegetation differ with species composition, elevation and bedrock types for species-rich tropical forests on Mt. Kinabalu, Borneo. We used eight forest monitoring plots established in 1995 at four altitudes (700, 1,700, 2,700 and 3,100 m above sea level) on two geological substrates (Si-rich acidic sedimentary and less Si-rich ultrabasic igneous rock), where tree species composition, abundance, biomass and litterfall had been monitored. For live leaves of 71 dominant tree species (total relative basal area > 60% in each plot) and leaf litter collected in traps, Si concentration was determined after alkaline extraction. Si availability in the upper 10 cm of mineral soil was determined as Si dissolved to water after shaking overnight. Tree species with high leaf Si concentrations occurred mostly in the lowest elevation plots. The community-mean Si concentration in live leaves, as well as Si concentration in leaf litter, decreased with increasing elevation. The estimated annual flux of leaf litter mass and Si also decreased with increasing elevation. Leaf and litter Si concentrations showed no difference between the two bedrock types without interaction with elevation. Due to large turnovers of species composition with elevation and bedrock types, most species occurred only in one plot. For 11 species that occurred at two or three plots, only one species showed a weakly significant difference in leaf Si concentration between bedrock types. Surface soil Si availability was greater at lower elevation plots and differed with bedrock types only at 1,700 m. This pattern was consistent with a hypothesis that Si input via litter in the form of plant opal, rather than bedrock types, should influence the soluble Si in the upper soil horizons. These results suggest that Si cycling by vegetation is more active in lower elevation forests regardless of bedrock types, most likely because Si accumulating species are more abundant in lowland tropical forests.

## 1 Introduction

Silicon (Si) is a beneficial element for many plant species (Ma and Takahasi, 2002; Cooke and Leishman, 2011), and as such, soil availability of Si is an important aspect of plant communities and ecosystems (Cooke and Leishman, 2012). Whereas the ultimate source of Si is the earth's crust and rocks, it is increasingly recognized that terrestrial Si dynamics is influenced by



plants as they uptake Si from soil water and return to the soil via litterfall (Bartoli, 1983; Lucas et al., 1993; Alexandre et al., 1997). Once absorbed by roots, Si accumulates in plant organs as opal phytoliths, which have higher solubility than Si in aluminosilicate in clay and quartz in sand (Fraysse et al., 2009). The study by Lucas et al. (1993) from a lowland Amazonian forest was the first that demonstrated that Si returned to topsoil via litterfall significantly enriched the upper soil horizon. The

importance of such Si cycled by vegetation is increasingly recognized in temperate forests as well (Struyf and Conley, 2012). Yet, far greater levels of Si cycling by vegetation may exist in lowland tropical forests (Lucas et al., 1993; Alexandre et al., 1997). Furthermore, it remains unknown how tree species may differ in their leaf Si accumulation and differentially influence biogeochemical cycling of Si in species-rich tropical forest ecosystems.

Recently developed conceptual models on biogeochemical Si cycles incorporate the role of terrestrial vegetation in

addition to lithospherical and limnological processes (e.g., Gérard et al., 2008). The bedrock containing siliceous minerals of variable weatherability is the source of Si containing soil particles, including aluminosilicate complex of secondary clay minerals. In the traditional pedological view, Si dissolved in soil water is mainly derived from rocks (lithogenic) and soils (pedogenic), which may exist at 0.1-0.6 mM concentrations as orthosilicic acid ($H_4SiO_4$) (Epstein, 1994). However, it is increasingly recognized that plants influence Si movement in the soil. Plants absorb orthosilicic acid (uncharged molecules)

along with water through roots, and transport it to the shoot, where it may form opal phytoliths ($SiO_2 \cdot nH_2O$) (Epstein, 1999). Since Si does not re-dissolve from opal phytoliths within a live plant, it accumulates as plant organs age, especially in leaves (as reported by Motomura et al., 2004 for a bamboo). When plant organs containing Si die, their decomposition releases Si to the soil. It is increasingly recognized that opal phytoliths in dead plant tissues significantly contribute to biogenic silica (BSi) in the soil, although siliceous shells of soil protozoans (Clarke, 2003), diatoms and sponges (Conley and Schelske, 1993)

may also contribute to the soil BSi pool in some soils. Through decomposition of leaf litter, BSi is incorporated in the topsoil and becomes a potential source of dissolved Si through dissolution to soil water (Bartoli, 1983).

Biological Si cycling rate via litterfall varies widely among vegetation types (0.5-4.9 g Si $m^{-2}$ $yr^{-1}$) (reviewed by Lucas, 2001), possibly because plant species differ in Si uptake patterns. Using active transport mechanisms in roots, some species reject Si uptake, while others actively concentrate Si into root xylems (Ma and Takahasi, 2002). Si concentrations in

leaves of 735 plant species from 125 studies compiled by Hodson et al. (2005) demonstrate that leaf Si concentration differs widely among phylogenetic groups. But, their analysis is strongly biased to temperate plant taxa, in which only six studies encompassing 27 species are tropical. For mechanistic understanding of Si cycling in a tropical forest, we need data on species composition and abundance, as well as Si uptake of dominant tree species.

Elevation is a factor that may significantly influence Si cycling, because temperature regimes, soil weathering rates,

plant productivity, and nutrient cycling patterns change with elevation. Forest structure and tree species composition also change with elevation. Forest productivity and species diversity decrease with increasing elevation from 700 m to 3,100 m on Mount Kinabalu (Aiba and Kitayama, 1999; Kitayama and Aiba, 2002). We predict that Si cycling rates should decline with elevation, not only because annual leaf litterfall mass declines leaf litter Si flux with increasing elevation, but also because tree species composition changes with elevation. On Mt. Kinabalu, lowland taxa such as Dipterocarpaceae dominate



at 700 m above sea level (asl), whereas Fagaceae and Myrtaceae dominate at 1,700 m asl, and Podocarpaceae (gymnosperm) and Myrtaceae dominate at 2,700 m asl and above (Aiba et al., 2002). As gymnosperms accumulate less amount of Si in leaves than angiosperms (Takahashi et al., 1981; Hodson et al., 2005; Piperno, 2006), the increasing dominance of gymnosperm trees with increasing elevation on Mt. Kinabalu may decrease community-wide concentrations Si in leaves.

Bedrock type may also influence forest Si cycling because bedrocks differ in amounts of various siliceous minerals and influence plant species composition via mineral nutrient availability. On Mt. Kinabalu, plant communities differ greatly on Si-rich acidic sedimentary rock (> 50% $SiO_2$) and less Si-rich ultrabasic igneous rock (41% $SiO_2$) (Jacobson, 1970). But, total Si contents of bedrocks may not indicate the rates of Si release from them, because weathering rates tend to be slower in quartz-rich sedimentary rocks than in ultrabasic rocks (Brooks, 1987). It is important to recognize Si concentration in

leaves may not be a simple function of Si solubility of bedrocks, because plant species differ in their Si uptake strategies. Active uptake may compensate for low-Si availability in the soil, whereas rejective-uptake species avoid accumulation of Si even in soils of high Si availability. Little is known about how plant communities of high species diversity, such as tropical forests, on different bedrocks may differ in their influences on Si cycling by vegetation.

      Here, we report how leaf Si accumulation and the community-wide estimate of annual Si flux via litterfall

change with elevation and bedrock types, taking advantage of the existing series of forest monitoring plots on Mt. Kinabalu (Aiba and Kitayama, 1999). These plots, located from 700 to 3,100 m asl on two contrasting bedrocks, offer an opportunity to explore how Si cycling by vegetation changes with tree species composition, elevation and bedrock types in tropical forest ecosystems. We evaluated three apriori predictions. Firstly, we predicted that annual Si flux via leaf litterfall should be smaller in higher elevation plant communities. Our second prediction was that Si cycling via vegetation may differ between

sedimentary and ultrabasic rocks that differ in Si contents and solubility. Thirdly, we predicted that litter Si flux should reflect the abundance-weighted leaf Si concentrations of dominant tree species. For species that occurred at multiple plots spanning a range of elevations and bedrock types, we examined within-species variation in leaf Si concentrations. We also compared water extractable Si concentration across the elevation range and bedrock types as an indicator of Si availability in the soil, and also as a possible correlate of variation in Si returned to the soil via leaf litter. Doing so, our study quantifies for

the first time in a forest ecosystem how Si flux varies in relation to species composition, elevation and bedrock types.

## 2 Methods

### 2.1 Study site

This study was conducted in the humid tropical montane forests on Mt. Kinabalu, Sabah, Malaysia (summit height 4,095 m

asl, 6°05'N, 160°33'E). Samples were collected from eight plots laid at four elevation levels (700, 1,700, 2,700 and 3,100 m asl) corresponding to four vegetation zones (hill dipterocarp, sub-montane, montane and sub-alpine) and two bedrock types (sedimentary rock with $SiO_2$ content > 50% and ultrabasic rock with $SiO_2$ content ≃ 41%). A slight deviation from this scheme was that the "sedimentary" rock site at 3,100 m elevation was actually on a granitic rock (> 60% $SiO_2$). The plot size varied from 1 ha to 0.06 ha (smaller size at higher elevation with lower forest statue near the tree line, see Table S1). In total,




380 species were recorded across the eight plots ($\geq$ 10 cm dbh) (Aiba and Kitayama, 1999). The climate was humid tropical with minor seasonal temperature fluctuations (Kitayama, 1992). The mean annual temperature ranges from 23.9ºC at 700 m asl to 10.6ºC at 3,100 m asl (calculated from Kitayama, 1992). The annual rainfall is approximately 2,300 mm yr$^{-1}$, with little change with the elevation (Kitayama and Aiba, 2002). See Aiba and Kitayama (1999) for further details of the sites and their

environments, and Wagai et al. (2008) for basic soil properties including clay mineralogy.

**2.2 Leaf and litter sampling**

Dominant species that together accounted for > 60% relative basal area (RBA) of canopy trees ($\geq$ 10 cm dbh) in each plot (Aiba et al., 2002) were sampled, resulting in a total of 71 species belonging to 28 families across the eight plots (Table S1 in

the Supplement). The majority of the 71 species occurred at a single plot, with 11 species occurring at more than one plot (8 and 3 species occurred at 2 and 3 plots, respectively; (Table S2 in the Supplement). Terminal branches were collected from a sun-lit portion of each crown of at least three individuals per species per plot with combination of slingshot and rope-climbing techniques from July to November, 2001. For three species, just a single individual was available for leaf collection. Samples were immediately transported to a field laboratory, where fully expanded and matured leaves were oven dried at

70˚C. The dried leaves, excluding petioles, were ground and stored in airtight plastic bags in air-conditioned laboratories until further analysis in the laboratory.

Litter was collected at two-week intervals in April, 1997 and May and August, 1998 from 20 traps (0.5 m$^2$ frames covered with a fine-meshed net in the shape of inverted cones) at 700 and 1,700 m elevation plots and from 10 same-sized traps at 2,700 and 3,100 m elevation plots. They were dried at 70°C and separated to leaves, branches, reproductive parts,

and dusts. Each type of litter samples were combined to make a pooled sample per plot before grinding for analysis of elemental concentrations. Ground samples were sealed in plastic bags and preserved in air conditioned laboratories. Preliminary analyses found little temporal variations in concentrations of nitrogen and phosphorus in leaf litter samples (Kitayama and Aiba, 2002). We report only Si concentration of leaf litter, which had far greater Si concentration and total mass compared to other types of litter. The leaf litter Si concentration was significantly correlated among different collection

times (Fig. S1 in the Supplement).

**2.3 Soil sampling**

Soil samples were collected in March 2015 from the upper mineral horizon (0-10 cm, largely corresponding to A-horizon) where most fine roots occur (Wagai et al., 2011). In each plot, five transects were set parallel to each other, along which five

cores were taken every 5 to 10 m using a stainless steel sampler (4.8 cm diameter) after removing O-horizon. The five cored samples were combined by transect and homogenized, and visible roots were removed by hand before air-drying and sieving (2 mm).

**2.4 Si extraction and measurements**



Dried leaf and leaf litter samples were milled to a fine powder. From ca. 30 mg of ground sample, Si was extracted with alkaline extraction of BSi with 1% sodium carbonate in water (20 ml in a polycarbonate bottle, shaking at 85°C overnight) in 2015, adopting the method of Conley and Schelske (1993). The Si concentration in the extract was immediately measured with the molybdate blue colourimetry (Sauer et al., 2006; Faisal et al., 2012).

5         For measurements of water extractable Si concentration, about 3 g of each air-dried soil sample was mixed in 20 ml of distilled water in a 50 ml centrifuge tube in 2015. The tube was shaken for 20 hours at room temperature (ca. 180 oscillations per minute), then centrifuged at 10,000 rpm (ca. 12,000 $g$) for one hour. The supernatant was suction-filtered through a 0.025 μm membrane filter, and 1 ml of the filtrate was immediately analyzed with the molybdate blue colourimetry. Other chemical and physical properties of the soil samples were measured at the Institute for Agro-

Environmental Sciences, Tsukuba, Japan.

### 2.5 Statistical analyses

For each plot, abundance-weighted community mean of the leaf Si concentration was calculated as follows Eq. (1):

Abundance-weighted leaf Si concentration $= \sum_{i=1}^{s} L_i \times \frac{RBA_i}{TRBA}$,                (1)

$L_i$ is mean leaf Si concentration of species i, $RBA_i$ is RBA of species i, and $TRBA$ is total RBA of all the target species at each plot (Aiba et al., 2002). Leaf litter Si flux was determined by multiplying leaf litter Si concentration by leaf litterfall dry mass (kg ha$^{-1}$ yr$^{-1}$) reported previously (Kitayama and Aiba, 2002). The effects of elevation (700, 1,700, 2,700 and 3,100 m asl) and bedrock types (sedimentary and ultrabasic) on species leaf, leaf litter and water extractable Si concentrations among the plots were tested with analysis of variance (ANOVA). Leaf Si concentration was log-transformed to improve normality

before ANOVA. Si concentration linearly increases with leaf age (Motomura et al., 2004). Because we were interested in estimating the maximum leaf Si concentration that would be found in old and senescing leaves, we used the 90th percentile of measured leaf Si concentrations as a species trait observed in each plot. For the 11 species that occurred at more than one plot, we used ANOVA to examine to test within-species variation after applying Bonferroni correction for multiple comparisons.

25         Phylogenetic independent contrasts (PICs; Felsenstein, 1985) were calculated for the mean leaf Si concentration and elevational distribution of the 71 species in the data set, using the software Phylomatic (version 3; Webb and Donoghue, 2005), the latest updates from the Angiosperm Phylogeny Group III (Bremer et al., 2009), and the ape package (Paradis et al., 2004). The observed patterns of PICs were compared with a null model assuming a unit branch length (except for polytomies, for which a negligibly small branch length was assumed) and the Brownian motion of evolutionary change, using the ape

package. Pearson's $r$ for PIC correlations was calculated between foliar Si concentration and elevation values (Garland et al., 1992). Blomberg's $K$ statistic was also calculated to examine the degree of phylogenetic signals (Blomberg et al., 2003), using the phytools package (version 0.5.64; Revell, 2012). All statistical analyses were performed in R 3.1.3 GUI 1.65 (R Development Core Team, 2016) with a significance level of alpha = 0.05.





## 3 Results

### 3.1 Si cycling via leaf litterfall

The annual leaf litter Si flux differed substantially in relation to elevation differences of the eight forest monitoring plots on Mt. Kinabalu, with little difference between the bedrock types (Fig. 1). The Si concentration per unit leaf-litter dry mass was the highest at 700 m elevation plots (6.0 and 5.0 mg Si g$^{-1}$ for sedimentary and ultrabasic rocks, respectively), and decreased with increasing elevation (Fig. 1a). Leaf-litter Si concentration strongly differed by elevation but not by bedrock type without a significant interaction between the two (Table 1). Annual leaf litterfall mass decreased with increasing elevation

and was higher on the sedimentary rock than the ultrabasic rock except at 2,700 m asl (reproduced from Kitayama and Aiba 2002, Fig. 1b). Consequently, annual Si flux via leaf litterfall (Fig. 1c) decreased with increasing elevation, without noticeable differences between the sedimentary and ultrabasic rocks, except at the highest two elevations on the sedimentary rock.

### 3.2 Variations of leaf Si concentration among species and communities

Mass-based leaf Si concentration differed substantially among the 71 tree species and the elevation levels (Fig. 2). The mean leaf Si concentration varied up to 57 folds among the species (0.24-13.6 mg Si g$^{-1}$) (Table S2 in the Supplement). The interspecific variation of leaf Si concentration within a plot was greater in the lower elevation plots. Species leaf Si concentration significantly differed by elevation but not by bedrock type without a significant interaction between the two

(Table 1). Abundance-weighted leaf Si concentration was the highest at 700 m asl on both bedrocks (Fig. 3).

        Because species composition changed almost completely from plot to plot (Table S1 in the Supplement), the observed among-plot differences in leaf Si concentration may reflect species traits, environmental responses of each species or both. Thus, we compared leaf Si concentrations for the 11 species that occurred at more than one plot, most of which occurred in the higher elevation plant communities (Table S2 in the Supplement). Of the 11 species, only *Leptospermum*

*recurvum* showed significant difference among three plots ($P < 0.005$, after applying the Bonferroni correction) (Fig. 4), showing that physiological responses of each species had very limited effects on observed differences among plots if any.

        The Blomberg's $K$ statistic showed a significant phylogenetic signal for elevation ($K = 0.37$, $P < 0.05$) (see Fig. S2 in the Supplement), reflecting the affinity of gymnosperms, Theaceae, Symplocaceae and oaks to the high elevation plots, and the affinity of the clade containing Dipterocarpaceae, Thymelaeaceae, Bombacaceae, and Sterculiaceae to the lowest

elevation. In contrast, phylogenetic signal was weaker and non-significant for leaf Si concentration ($K = 0.27$) (see Fig. S3 in the Supplement). PIC values of elevation explained 15% of the variation in PIC values of leaf Si concentration ($r = -0.39$, $P < 0.001$), suggesting that evolutionary shifts to a higher elevation are associated with less degrees of leaf Si accumulation.

### 3.3 Si bioavailability in soil



Water extractable Si concentration per unit soil mass differed among elevations and between the bedrock types with a significant interaction between elevation and bedrock (Table 1). The mean water extractable Si concentration at 700 m elevation plots (0.021 mg Si g$^{-1}$ for both bedrock types) was much higher compared to other elevations (0.007-0.015 mg Si g$^{-1}$) (Fig. 5). The Si value differed by bedrock only at 1,700 m elevation.

## 4 Discussion

We predicted that litter Si flux should differ with elevation and bedrock types, in association with differences in leaf Si concentrations of dominant tree species and soil Si availability. The results supported this prediction in terms of elevation, but we found no significant differences in litter or leaf Si concentrations between the two bedrock types. These results likely

reflected proximate environmental factors, such as cooler temperature regimes at higher elevation, but more importantly, species differences that result in difference in leaf Si concentrations in relation to elevation.

The annual leaf litter Si flux was greater at lower elevation plant communities, with large difference between 700 m and 1,700 m asl. We suspect that Si cycling rates by vegetation would be even faster in lowland forests below 700 m asl due to even greater forest productivity and high diversity of angiosperm trees. The published data on Si flux via leaf litter

measured in lowland tropical forests (lowland Amazonia, Lucas et al., 1993; Congo, Alexandre et al., 1997) show even higher values (3.3-7.6 g Si m$^{-2}$ yr$^{-1}$) than our 700 m elevation plots (3.1-4.4 g Si m$^{-2}$ yr$^{-1}$). We are not aware of published litter flux data from other lowland tropical forests, but unpublished data for lowland dipterocarp forests from our research group show high values (e.g., 8.5 g Si m$^{-2}$ yr$^{-1}$ for Pasoh, Peninsula Malaysia, per undergraduate thesis of Hidehiro Ishizawa, Kyoto University, and similar values for lowland Borneo sites). These litter Si flux values for tropical lowland forests are

high compared to European forests (ranging from 0.2 g Si m$^{-2}$ yr$^{-1}$ for a black pine stand to 4.2 g Si m$^{-2}$ yr$^{-1}$ for a Norway spruce stand) (Cornelis et al., 2010).

The higher litter Si flux at lower elevation sites reflected both greater productivity and high litter Si concentrations. The latter, in turn, reflected higher community-level means of live leaf Si concentrations in lower elevation plots. Whilst the turnover of dominant tree species with elevation most likely explains the community-wide trend of foliar Si concentrations

on Mt. Kinabalu, we cannot completely rule out other possible explanations. Firstly, all else being equal, warmer climate at lower elevation may mean greater transpiration and faster rate of Si transport to leaves with xylem water. With cucumber, which is a moderate active uptaker of Si, experimental manipulation of transpiration results in different levels of Si accumulation (Faisal et al., 2012). At the same time, this and other studies with herbaceous species (e.g., Liang et al., 2006) show that Si accumulation rate is not merely transpiration-dependent, but also modulated by active uptake by roots, such that

high Si accumulating species compensate for low Si availability from the soil. To isolate the physiological effect of transpiration rates on Si accumulation at different elevations, a reciprocal transplanting of high and low elevation species should be conducted in the future.

The second temperature-related abiotic factor that might increase leaf and litter Si concentration is the faster weathering of silicate minerals under warmer climate (Wagai et al., 2008). Faster weathering enhances the pool size of



soluble Si, via its direct contribution to dissolved Si concentration, which may be absorbed by plants and returned to the soil as BSi contained in dead plant tissues (Cornelis and Delvaux, 2016). As BSi is more soluble than Si contained in rock and clay (Fraysse et al., 2009), a positive feedback to accelerate Si cycling by vegetation may develop. While we cannot totally discount such direct and indirect effects of temperature regime on the observed elevational trends of leaf and litter Si

concentrations, within-species variations for species that occurred at multiple elevations (Fig. 4) favor biological factors over abiotic and proximate environmental factors.

A likely biological explanation for the high leaf Si concentrations in lowland trees is an evolutionary one. Strong phylogenetic signals exist for Si accumulation among a broad range of land plant species (Takahasi et al., 1981; Hodson et al., 2005) even though the signal was weak within our 71 species dataset, and high Si accumulating species up-regulate

active uptake mechanisms (Ma and Takahashi, 2002; Liang et al., 2006). In other words, each plant regulates Si uptake rates as a species-specific trait. A key trend found in our study was that tree species with high leaf Si concentrations occurred more in lower elevation forests. This suggests a tantalizing possibility that active Si uptake is adaptive in relation to biotic and abiotic stress factors in lowland tropical forests compared to forests of higher altitudes and latitudes. Grasses and other herbaceous species are known to use silica to enhance defense against herbivory and disease (McNaughton and Terrants,

1983; Ma and Takahashi, 2002; Hartley et al., 2015). While such defensive use of silica has been reported only sporadically for tropical woody plants (e.g., Korndorfer and Del Claro, 2006), it is worth testing with many other tropical trees which are exposed to a diverse array of herbivores and disease agents year around. Another potential stress factor in lowland tropical forests is a high risk of aluminum toxicity, especially where soils are highly weathered and acidic. It is possible that some lowland tropical tree species have evolved Si accumulation as part of their strategies to ameliorate aluminum and heavy

metal toxicity as reported in some temperate herbaceous and tree species (Epstein, 1994; Hodson and Sangster, 1999). As these are speculative ideas, we need more data from tropical taxa about their Si uptake patterns, as well as experiments to test these possible benefits of Si accumulation.

Our study found no differences in leaf and litter Si concentrations between the two bedrock types contrasting in total Si contents, which may suggest that plant Si uptake traits override the effect of bedrock. Shaller et al. (2018) also report

intraspecific variation in foliar Si content that does not respond to soil Si availability in Panamanian tropical forests. In our study, species composition was almost totally different between the two bedrock types. Similarly to our results, community-wide leaf Si concentration does not depend on Si availability differences between contrasting soil types in Australia (Cooke and Leishman, 2012). These results strongly suggest that species traits influence biogeochemical cycles of Si more than proximate environmental factors. But, we cannot rule out a possibility that leaf and litter Si concentrations did not differ

between the bedrock types simply because these rocks did not differ in weatherability to release Si into more soluble forms in the soil. Sedimentary rocks contain more Si than ultrabasic rocks (Brooks, 1987), but abundance of quartz in some sedimentary rocks (such as sandstones) may mean slower Si dissolution from such rocks than from ultrabasic rocks (Goldich, 1938; Schulz and White, 1999). Hence, we need to integrate mineralogical processes and biological processes. We quantified





Si availability only for a shallow part of the A-horizon, but future studies need to evaluate Si availability from greater soil depths where trees may also obtain water, while simultaneously distinguishing biogenic *vs.* rock-derived Si.

In our dataset, it is intriguing that lowland trees in the family Dipterocarpaceae differed substantially in leaf Si concentrations, from 0.46 mg Si g$^{-1}$ (*Shorea parvistipulata*) to 9.87 mg Si g$^{-1}$ (*Shorea gibbosa*). Dipterocarpaceae represents

the major group of dominant tall trees in lowland tropical forests in Asia (Richards, 1952; Ashton, 2015), and as such, further investigation on their Si uptake patterns can reveal significant insights on Si cycling by vegetation in this region. In addition, Dipterocarp-dominated forests have been severely logged during recent decades (Ashton, 2015), and this may have strongly impacted biogeochemical cycles of Si in the tropics. Conley et al. (2008) suggest that conversion of forests to agricultural and open lands cause significant losses of DSi from the soil and may adversely affect temperate terrestrial

ecosystems. The potential impacts of deforestation and forest degradation on Si cycling may be even greater in the tropics given the abundance of high Si accumulating species in lowland tropical forests. In order to analyze these potential impacts over longer time spans, quantification of plant parts other than leaves (e.g., coarse woody debris and tree roots) that also contribute to forest Si cycling will be needed (Clymans et al., 2016; Turpault et al., 2018).

Greater understanding of both proximate and ultimate factors that regulate biogeochemical cycles of Si in tropical

ecosystems is urgently needed, including how species diversity matters for Si cycle by tropical vegetation. Our study has scratched just the surface of this topic.

**Data availability**

All data used in this manuscript will be available in the Dryad Digital Repository.


**Authors' contribution**

K. Kitajima conceived the ideas; K. Kitajima, K. Kitayama and R. Nakamura designed methodology; R. Wagai, S. Suzuki and K. Kitayama collected samples; R. Nakamura and H. Ishizawa performed the chemical analyses; K. Kitayama and R. Wagai provided critical information on the forest monitoring plots on Mt. Kinabalu; R. Nakamura and K. Kitajima led data

analyses; R. Nakamura wrote the first draft of the manuscript. All authors contributed critically to the drafts and gave final approval for publication.

**Competing interests**

The authors declare that they have no conflict of interest.


**Acknowledgements**

This research was supported by the grants 26650163, 22255002 and 26660051 from the Japan Society for the Promotion of Science. We thank the Sabah Parks for permission and logistic support for research on Mt. Kinabalu. Soil import followed





the regulations specified by a permit issued by the Ministry of Agriculture, Forestry and Fishery in Japan. We also thank
Joseph Phillips for English language editing.

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





**Table 1.** ANOVA results for the effects of elevation (700, 1,700, 2,700 and 3,100 m asl) and bedrock type (sedimentary and ultrabasic) on species mean leaf Si concentration (log-transformed for normalization), leaf litter Si concentration and water extractable Si concentration on a basis of sample dry mass. Species leaf Si concentration was defined as the maximum

5   capacity to accumulate Si in leaves, and represented by 90th percentile of measured values by species in this study. In the case the species occurred at multiple plots, species leaf concentration was determined by plot.

| | Elevation | | | Bedrock type | | | Interaction | | |
|---|---|---|---|---|---|---|---|---|---|
| | df | F | P | df | F | P | df | F | P |
| $Log_{10}$ (Species leaf Si concentration ($mg\ g^{-1}$)) | 3 | 8.74 | **< 0.001** | 1 | 0.7 | 0.41 | 3 | 0.97 | 0.41 |
| Leaf litter Si concentration ($mg\ g^{-1}$) | 3 | 20.4 | **< 0.001** | 1 | 0.36 | 0.56 | 3 | 0.35 | 0.79 |
| Soil water extractable Si ($mg\ g^{-1}$) | 3 | 22.1 | **< 0.001** | 1 | 5.0 | **0.032** | 3 | 3.2 | **0.036** |

*Notes:* Significant terms ($P < 0.05$) are shown in bold. Leaf Si concentration had at least three replicates per species but *Shorea parvistipulata*, No.14 on ultrabasic rock, *Tristania* sp.2, No.49 and *Heritiera simplicifolia*, No.64 in Table S2 (n = 1). Leaf litter and water extractable Si concentrations had three and five replicates, respectively.

20





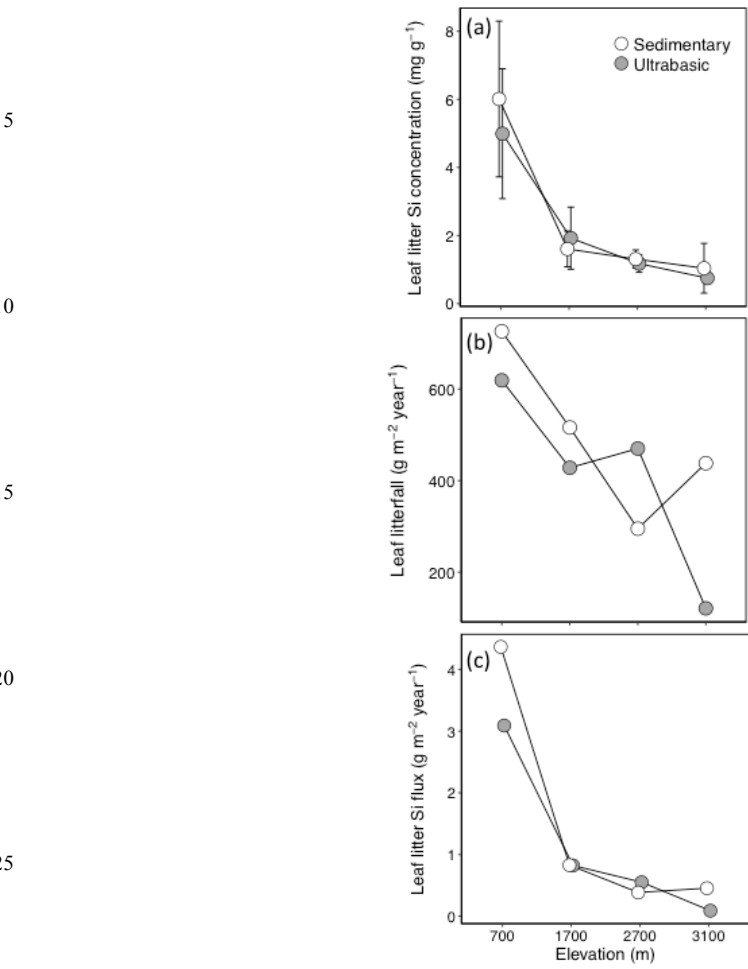

**Figure 1.** (a) Leaf litter Si concentration (means ± s.d. for three two-week-interval collections), (b) annual leaf litter mass
(data from Kitayama and Aiba (2002), showing the mean from two years of data), and (c) leaf litter Si flux (n = 1) at
different elevations for sedimentary (white circles) and ultrabasic (grey circles) rocks.

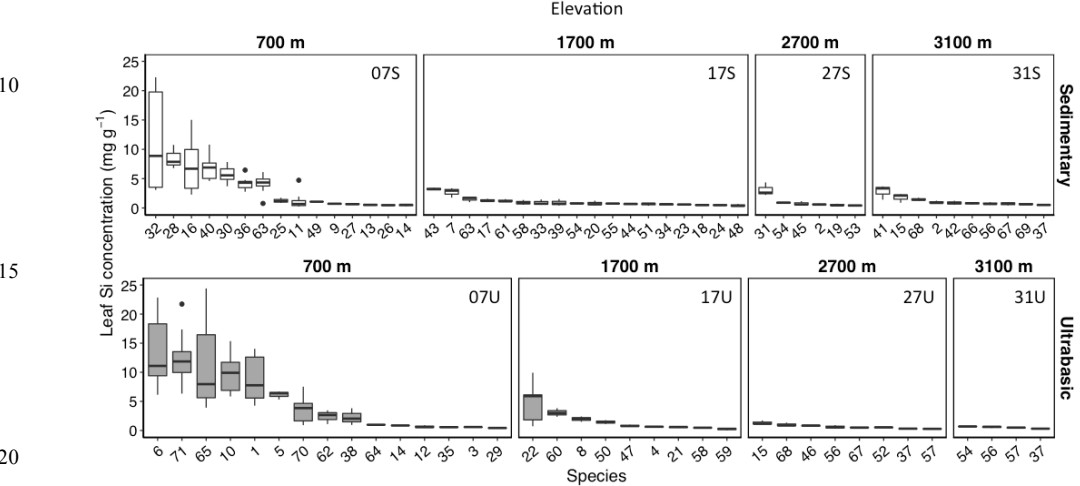

**Figure 2.** Variations of leaf Si concentrations among species and plots, showing the species median (thick horizontal line) with 25-75$^{th}$ percentiles (box) and 95% confidence interval (whiskers) for each species. Species identity is indicated by the number on the horizontal axis (see Table S1 in the Supplement). There was a large species turn-over among the eight plots located at different elevations on two types of bedrock. Open and shaded boxes indicate sedimentary and ultrabasic rocks (sedimentary and ultrabasic rocks, upper and lower panels, respectively). Eleven of the 71 species occurred at multiple plots and are shown at more than one panel (8 species at two plots, 3 species at three plots, see Fig. 4 and Table S2 in the Supplement). Plot ID is indicated in each panel.

30





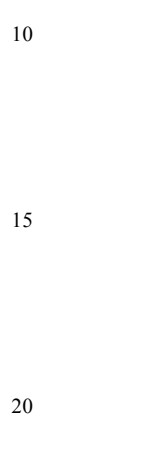
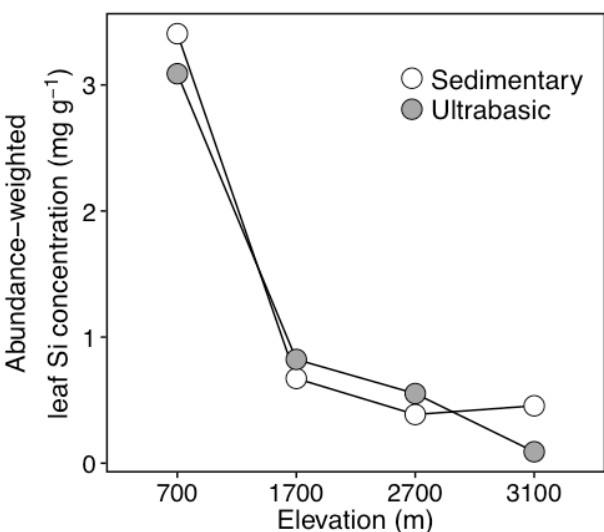

**Figure 3.** Abundance-weighted leaf Si concentration (n = 1) at different elevations for sedimentary (white circles) and ultrabasic (grey circles) rocks. Each point represents a plot at the elevation.





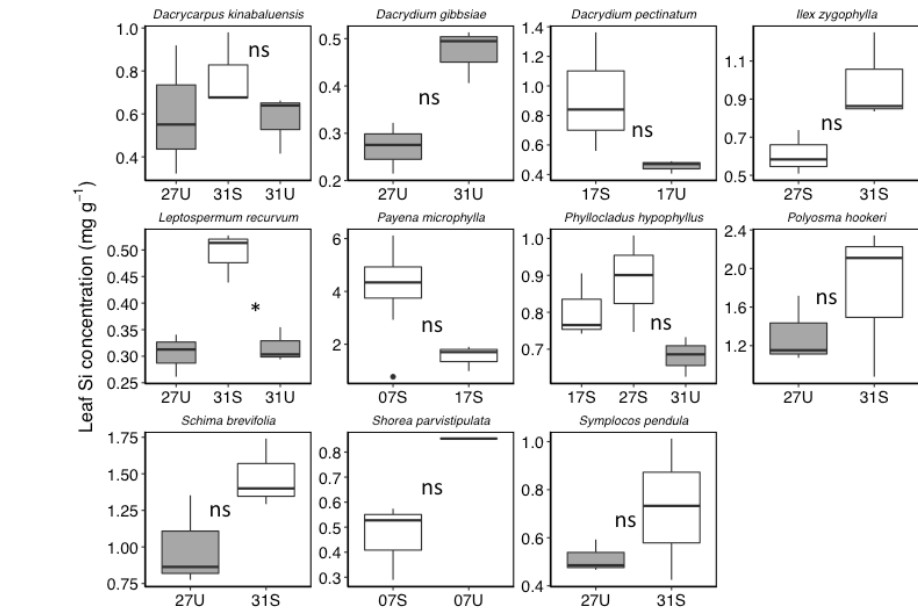

**Figure 4.** Boxplots for leaf Si concentration for 11 tree species that occurred at multiple plots. Open and shaded boxes indicate sedimentary and ultrabasic rocks. See Fig. 2 for Plot ID. Significant site effect was found for one species * ($P <$ 0.0045 with Bonferroni correction) by ANOVA.



20

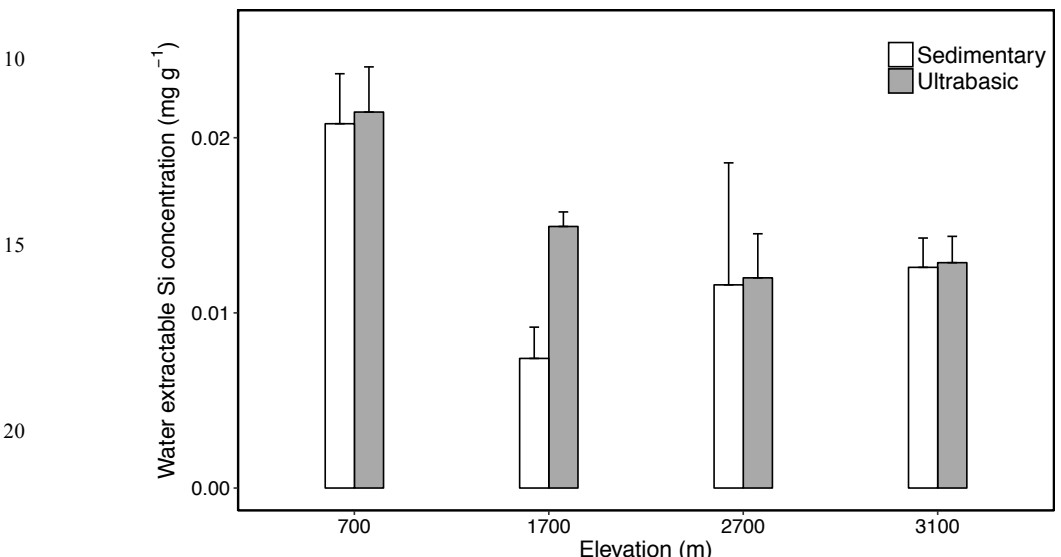

**Figure 5.** Water extractable Si concentration per unit dry mass of soil for the plots at specified elevations on sedimentary
(white bars) and ultrabasic (grey bars) rocks. Data are means + s.d. (n = 5).

30