# Peer review of "Silicon cycled by tropical forest trees: effects of species, elevation and bedrock on Mount Kinabalu, Malaysia"

_Biogeosciences, 2018_

## Referee Comment (RC1) · Anonymous Referee #1 · 17 Dec 2018

Main comments : This paper brings some new informations regarding the Si content of different tropical species, on the Si concentration in litterfall in eight plots sampled at three dates and on Si concentration in soil (0-10 cm depth horizon) obtained through water extraction, in these eight plots for one date. These 8 plots are located at 4 different elevations (700, 1700, 2700 and 3100 m). For each elevation plots are placed on different bedrocks, i.e., acidic sediment and ultrabasic igneous rock; except for 3100 m where sediment bedrock was replaced by granite. These situations lead to strong differences between ecosystems (structure, vegetation composition, production rate, littefall, litter decomposition, soil fertility. . .) previously published in Kitayama and Aiba 2002. For examples, wood biomass ranged from 3.5 to 43.3 Kg.m-2, leaf biomass

varied between 0.17 to 0.56 kg.m-2 and leaf litterfall between 121 to 1113 g.m-2.yr-1, litter decomposition constant ranged between 0.44 and 1.69 yr-1 and pH water ranged from 3.4 to 5.4. These data from Kitayama and Aiba (2002) in a tropical ecosystem are exceptional. So you have different ecosystems with very contrasting production and recycling rates (as well as floristic composition).

INTRODUCTION I am not convinced by the hypotheses presented in the introduction which are based on indirect factors. The total Si concentration of a rock type is not a determining factor of the flux of Si in the soil and the soil can act indirectly via its fertility on the biological cycle of the Si. The elevation is an indirect factor which can affect various components of the recycling of an element via the production and the decomposition of litters notably. To use indirect variables strongly limits the interpretations. Beyond elevation and rock type, the differences of production and of litter decomposition rate of these 8 ecosystems seem to be fundamental and direct factors affecting the concentrations of Si in leaves. The hypothesis of this article could be for example that the influence of the recycling rate of Si by the stand on the Si concentration in leaves in the long term. See the paper of Cornelis and Delvaux (2016) which is cited in this manuscript.

METHOD Tropical ecosystems are complex. In consequence, you have to well describe the ecosystems (see Tab 1 Kitayama and Aiba 2002), with matter stocks and fluxes. Specify whether they are affected by forest management and understory vegetation. Describe soil and their properties. Justify that you measures only Si concentrations in the samples collected at three dates (April 1997, May and August 1998) and during two weeks while the annual concentration was aimed. What are the limits of detection and uncertainties of our measurement devices? Soil description was absence.

RESULTS The original results of the Si concentrations in leaves are not enough valorized (except figure 2 which is a global figure). For example add the values of Si concentration for each species in the table S1.

DISCUSSION I suggest to rework a part of the discussion by dealing with the relations between direct factors such as litter production, litter decomposition, Si recycling, and Si concentration in leaves.

In conclusion, in the current state, this manuscript is not publishable in Biogeosciences because the research question is not enough relevant and not supported by an adequate device of observation allowing to conclude (three factors: species, elevation, bedrock and 2 indirect factors on the cycle of Si). However, the set of data stemming from this article could be better exploited by associating the data on the production and the decomposition of litters published in Kitayama and Aiba (2002).

---

## Referee Comment (RC2) · Anonymous Referee #2 · 10 Jan 2019

This paper presents a thorough study of the Si concentrations in multiple tree species across an elevation gradient in a tropical forest. It also calculates the annual litterfall associated Si fluxes, based on annual litterfall fluxes and Si concentrations. This is related to Si availability in the soil water.

The authors have done a great job in analysing a large set of tropical trees for Si concentration, and the consequent calculation of associated tree litter Si fluxes is interesting. The study has a strong merit for that, as it is one of the first –and probably the first at this scale – to analyse this potentially interesting component of tropical forest Si cycling.

That said, in my opinion, the study cannot deliver any strong new information on the effect of elevation or bedrock on the Si cycling by trees. My main concern is that the Si concentration in the leaves, and the associated Si litter flux, could just be a reflection of the Si availability in the soil water. The latter is highest at lowest elevation, and this results in generally higher Si concentrations in biomass. It is well known that all species, even those not accumulating Si but acquiring it passively, will show a higher Si concentration in biomass when availability is higher. The question thus is: why is Si availability higher at lowest elevation? Is it because of higher Biogenic Si concentrations in the soil? Is it because of alterations in soil water source?

The authors cannot provide a response to that with this analysis or dataset. In order to move this study from an interesting local observation, to a study that truly moves knowledge on Si cycling in tropical forest sites, a lot of extra context is needed, e.g. -How much BSi is present in the soils? -Indication of Si concentrations in porewater and Si leaching / import from / to groundwater. -Decomposition experiments with the litter.

Now, it is impossible to assess what the observed changes in Si concentrations in leaves, and associated litterfall Si fluxes, mean in the context of ecosystem Si cycling.

The dataset has large merit in itself, since it is quite unique in analysing tropical tree litterfall Si fluxes in such detail. However, it is difficult to currently make any strong conclusion on interaction between tropical trees and tropical Si cycling based on this study alone. For that, more information on soil Si pools and Si leaching / Si groundwater input is needed.

Still, as a scientist interested in Si cycling, I enjoyed reading this paper and am puzzled by its results. I am therefore hesitant to just indicate that the paper should not be published. Indeed, this paper could be a trigger for future work on the Si cycling in tropical forests. I would therefore recommend that, if possible, authors include some of the suggested information on the local Si biogeochemistry.

If that is not available, I would recommend authors to focus more on linking plant traits to Si concentrations observed. Is that different between elevations? Can you find an explanation for the large observed differences between species? I think the data are much more suited for that purpose, than for trying to put it in a larger biogeochemical context.

―――――――――――――――――――

---

## Author Comment (AC1) · 30 Jan 2019

Please find the attached PDF file for the detailed response to your comments and our revised manuscript.

Please also note the supplement to this comment:
https://www.biogeosciences-discuss.net/bg-2018-447/bg-2018-447-AC1-supplement.pdf